# Effects of Resveratrol-Loaded Cyclodextrin on the Quality Characteristics of Ram Spermatozoa Following Cryopreservation

**DOI:** 10.3390/ani14182745

**Published:** 2024-09-23

**Authors:** Ahmet Eser, Selin Yağcıoğlu, Ramazan Arıcı, Kamber Demir, Kemal Ak

**Affiliations:** 1Department of Reproduction and Artificial Insemination, Faculty of Veterinary Medicine, Siirt University, Siirt TR-56100, Turkey; 2Department of Reproduction and Artificial Insemination, Faculty of Veterinary Medicine, Istanbul University-Cerrahpasa, Istanbul TR-34320, Turkey; selin.yagcioglu@iuc.edu.tr (S.Y.); ramazan.arici@iuc.edu.tr (R.A.); kamberdemir@iuc.edu.tr (K.D.); kemalak@iuc.edu.tr (K.A.)

**Keywords:** semen freezing, resveratrol, sperm quality assessments, antioxidant

## Abstract

**Simple Summary:**

This study evaluated the impact of resveratrol-loaded cyclodextrin (RLC) on the quality of ram sperm after thawing. Our hypothesis was that resveratrol-loaded cyclodextrin, which enhances the antioxidant activity of resveratrol (RES), would improve the outcomes of freezing ram semen. Accordingly, this study investigates the effects of resveratrol at varying doses (10, 20, and 40 µg/mL), either in its pure form or loaded into methyl-β-cyclodextrin (CD), on sperm functions following the thawing of ram semen. The findings indicate that while cyclodextrin negatively affected sperm quality, particularly by increasing early capacitation and mitochondrial activity, resveratrol at a dose of 20 µg/mL effectively prevented early capacitation. Overall, both resveratrol and resveratrol-loaded cyclodextrin were generally effective and beneficial in the context of freezing ram semen.

**Abstract:**

This study investigated the effects of pure and methyl-β-cyclodextrin loaded forms of resveratrol (10 µg/mL, 20 µg/mL, and 40 µg/mL) on ram sperm functions post-thawing. Semen samples were pooled and divided into ten groups: Control, RES10, RES20, RES40, CD10, CD20, CD40, RLC10, RLC20, and RLC40. The groups were pre-diluted with media containing the group-specific chemicals, followed by 15 min of incubation, dilution, and freezing. To assess the effects of the chemicals, a post-thaw sperm quality assessment was conducted. Motility and other velocity parameters were evaluated using computer-assisted semen analysis. The functional integrity of spermatozoa membranes was assessed with the hypo-osmotic swelling test, and the capacitation status of spermatozoa was determined through fluorescent microscopic evaluation. Additionally, flow cytometry was used to evaluate mitochondrial activity, oxidative stress, and the integrity of the sperm membrane and acrosome. The results indicated that cyclodextrin adversely affected sperm functions following freezing–thawing, notably increasing the rate of spermatozoa exhibiting pre-capacitation and mitochondrial activity by approximately 34% and 16%, respectively (*p* < 0.05). It was found that 20 µg/mL resveratrol prevented pre-capacitation (*p* < 0.05). Both resveratrol and resveratrol-loaded cyclodextrin groups improved post-thaw sperm qualities overall, demonstrating their utility for freezing ram semen. However, higher concentrations of resveratrol were found to negatively impact sperm functions.

## 1. Introduction

Ram spermatozoa have a membrane structure that is proportionately rich in unsaturated fatty acids, but it is quite low in cholesterol. This causes ram spermatozoa to be more sensitive to the harmful effects of freezing and thawing [1]. In addition, membrane polyunsaturated fatty acids are the primary target of reactive oxygen species (ROS) attacks, which increases lipid peroxidation, especially during freezing and thawing, and causes motility, acrosome, and membrane damage in spermatozoa, thus reducing fertilization rates [2].

The procedures used in semen cryopreservation and the composition of the diluent are critical factors affecting post-thaw success [3]. To date, numerous species-specific procedures for dilution, cooling, freezing, and thawing, as well as various types of diluents, have been developed for successful cryopreservation. The effectiveness of different compounds added to the diluent has also been examined [4]. Researchers have aimed to reduce oxidative stress damage in cells by adding various endogenous and exogenous antioxidant compounds to semen extenders. Recent studies indicate that resveratrol, known for its antioxidant properties, added to semen extenders in various species, results in improved sperm functions in cooled or thawed semen [5,6,7,8,9,10,11,12,13,14,15,16,17,18,19,20,21,22,23,24,25,26,27]. However, there is insufficient research on the effectiveness of resveratrol in freezing ram sperm [28].

Resveratrol (trans-resveratrol; trans-3,4,5-trihydroxy-trans-stilbene) is a molecular compound in the stilbene family, comprising a methyl bridge and two aromatic rings. It protects plants against microbial and fungal diseases, ultraviolet radiation, and heavy metals [29]. Resveratrol (RES) possesses antioxidant properties due to its ability to scavenge hydroxyl and superoxide radicals. It is an extremely potent antioxidant in vivo, but its ability to capture reactive oxygen species is limited in vitro [30]. Resveratrol undergoes structural changes under room conditions due to interactions with oxygen, fluctuations in temperature, and exposure to light. Additionally, its solubility in water is quite limited [31]. Strategies for overcoming the constraints of resveratrol solubility are presently under investigation [32]. The drawbacks of resveratrol can be alleviated by incorporating it into the cyclodextrin (CD) molecule. CDs are cyclic oligosaccharides consisting of α (1→4)-linked D-glucopyranose subunits. The most common cyclodextrins are α-, β-, and γ-CDs, which contain six, seven, and eight glucose units, respectively. These are typically produced through the enzymatic degradation of starch. CDs possess a truncated cone shape, with hydroxyl groups oriented towards the outer surface of the cavity. This three-dimensional conformation results in a hydrophilic exterior and a hydrophobic interior cavity. This structural arrangement gives CDs water solubility and the ability to encapsulate a range of hydrophobic organic or inorganic molecules of appropriate size within their cavities. As a result, CDs form inclusion complexes that increase the uniform solubility of these molecules in water [33,34]. This property of CD also extends to resveratrol, enhancing its solubility, stability, and antioxidant capacity [35].

In this study, our hypothesis is that resveratrol-loaded cyclodextrin (RLC), which enhances the antioxidant activity of resveratrol, will increase the freezing success of ram sperm. To our knowledge, there is no existing study in the literature demonstrating the efficacy of RLC in ram sperm cryopreservation. Therefore, this study investigates the impact of resveratrol at varying doses (10, 20, and 40 µg/mL), either in pure form or loaded into methyl-β-cyclodextrin, on ram sperm quality characteristics following cryopreservation.

## 2. Materials and Methods

This study was conducted outside the breeding season (January–May) using Kivircik rams (*n* = 5) between two and five years old, housed at Istanbul University-Cerrahpasa Faculty of Veterinary Medicine (28° S, 41° W). Throughout the study period, the rams received standard management and feeding procedures and were housed separately from other animals.

### 2.1. Ethical Statement

The permission for this study was approved by Istanbul University Animal Experiments Local Ethics Committee (12.09.2018-56090).

### 2.2. Preparation of Semen Extenders

All of the chemicals used in this study were purchased from Sigma Chemical Co. (Saint Louis, MO, USA).

#### 2.2.1. Resveratrol-Loaded Cyclodextrin (RLC) Preparation

Commercial Methyl-β-cyclodextrin (C4555, Sigma-Aldrich, Inc. St. Louis, MO, USA), appropriate for use in cell studies, was employed in this study [36]. The resveratrol–cyclodextrin loading protocol was modified based on studies that analyzed conducted solubility, antioxidant properties, and photostability of the resveratrol–cyclodextrin complex [37,38]. First, 178.65 mg of methyl-β-cyclodextrin was dissolved in a glass test tube by adding 1 mL of deionized water. Next, 10 mg of resveratrol (R5010, Sigma-Aldrich, Inc. St. Louis, MO, USA) was added to this mixture. The resulting solution was mixed at 0.887 g in an orbital shaker (Solaris 2000, Thermo Fisher Scientific, Waltham, MA, USA) under room temperature conditions for 24 h with the lid closed. To prevent light exposure, the tube was covered with aluminum foil. After the mixing period, the mixture was transferred to a glass petri dish, frozen at −50 °C, and lyophilized at 0.203 Torr atmospheric pressure using a lyophilizer (Telstar, Lyo Quest, Telstar, Terrassa, Spain).

#### 2.2.2. Preparation of Tris-Citric Acid-Glucose (TCG) Medium

TCG medium (Tris 25 mM/mL, citric acid 8 mM/mL, and glucose 7 mM/mL) was employed to dissolve resveratrol, cyclodextrin, and resveratrol-loaded cyclodextrin substances added to the study groups for pre-dilution.

#### 2.2.3. Preparation of Tris-Based Egg Yolk (TEY) Extender

Tris-based egg yolk medium (Tris 27.1 g/L, fructose 10 g/L, citric acid 14 g/L, egg yolk 15%, pH 6.8–6.9, and 305–315 mOsm/kg) was used as the base sperm extender in all groups [39].

### 2.3. Experimental Design and Cryopreservation Protocol

The ejaculates were collected from rams using an electro-ejaculator (P-T Electronics, Model 302, Boring, OR, USA). Each ram’s ejaculate was individually analyzed for spermatological parameters using a computer-assisted semen analysis system, version 12.3 (CASA 12.3 IVOS, Hamilton-Thorne Biosciences, Beverly, MA, USA). Only samples meeting specific quality criteria (motility ≥ 80%; sperm concentration ≥ 1 × 10^9^/mL) were pooled (1 mL from each ram’s ejaculate) to minimize individual variations [39]. Afterwards, ten experimental groups containing 1 × 10^9^ motile sperm were formed (Table 1). In determining the preferred RES doses in this study, the study of Silva et al. [26] was taken into account. In addition, a 40 µg/mL RES group was added to determine the effect of high dosage on sperm functions.

Pre-dilution procedures with TCG (200 µL) containing group-specific chemicals were carried out in a water bath at 26 °C. An equal volume of chemical-free TCG was added to the control group to prevent dilution rate difference between the groups. Following pre-dilution, all groups were incubated for 15 min at 26 °C. Subsequently, each group was diluted with TEY solution (containing 5% glycerol, *v*/*v*), and the temperature was lowered to +5 °C at a controlled rate of 0.2 °C/min using a controlled-rate freezer (Bio-cool III, FTS Systems Inc., Stone Ridge, NY, USA). The samples were then maintained at this temperature for one hour to achieve equilibration [1,39]. Finally, the semen samples were frozen in liquid nitrogen vapor for ten minutes on a platform positioned 4 cm above the liquid nitrogen level [1]. For each group, 20 mini straws (0.25 mL) were stored in a liquid nitrogen container, and 50 × 10^6^ motile spermatozoa were loaded into each straw.

### 2.4. Sperm Quality Assessments

#### 2.4.1. Computer-Assisted Semen Analysis (CASA)

The CASA examinations, including total motility (%), progressive motility (%), velocity average path (VAP, µm/s), velocity straight linear (VSL, µm/s), velocity curve linear (VCL, µm/s), amplitude of lateral head displacement (ALH, µm/s), beat cross frequency (BCF, Hz), straightness (STR, VSL/VAP, %), and linearity (LIN, VSL/VCL, %), were conducted immediately after the equilibration was completed and after the thawing periods by analyzing approximately 600–800 sperm [40].

#### 2.4.2. Hypo-Osmotic Swelling Test (HOST)

The functional integrity of spermatozoa membranes was evaluated using the modified hypo-osmotic swelling test, as described by Bacinoglu et al. [41] and Cirit et al. [39]. Spermatozoa exhibiting twisting, folding, and/or spiraling in their tail sections under hypo-osmotic conditions were considered to have maintained functional membrane integrity.

#### 2.4.3. Assessment of Capacitation Status

Fluorescent chlortetracycline (CTC) dye was utilized to assess the capacitation status of spermatozoa following thawing. The staining method described by Perez et al. [42] was employed for this purpose. Stained spermatozoa were examined under a fluorescent microscope (Eclipse Ni-U, Nikon, Tokyo, Japan) at 400× magnification. The percentage of spermatozoa exhibiting B pattern (capacitated, fluorescence outside the post-acrosomal region) and F pattern (non-capacitated, entire head stained) statuses were determined by evaluating 100 cells with fluorescent reflection using 485/20 nm excitation and 580–630 nm emission filters for each experimental group (Figure 1).

#### 2.4.4. Flow Cytometric Analyses of Sperm Acrosome Integrity, Sperm Membrane Integrity, Mitochondrial Functions, and Oxidative Stress Status

Flow cytometry (Guava EasyCte™, Guava^®^ Technologies, Hayward, CA, USA) analyses were performed by evaluating at least 5000 spermatozoa. Data plot graphs of the analyses are shown in Figure 2.

FITC-PNA (fluorescein isothiocyanate labeled peanut agglutinin) and PI (propidium iodide) dyes were used together to assess acrosome integrity in spermatozoa after thawing. The staining technique described by Marco-Jimenez et al. [43] was used. The proportion of spermatozoa exhibiting green and/or red fluorescence excitation in the emission range of 519–590 nm was measured using flow cytometry. Spermatozoa that did not exhibit green and red fluorescence excitation (FITC-PNA and PI negative) were considered to be viable and have intact acrosomes.

The percentage of viable and membrane-intact sperm was assessed using flow cytometry with the combined application of the fluorescent dyes carboxyfluorescein diacetate (CFDA) and propidium iodide (PI). The method reported by Camara et al. [44] was used for the staining protocol. Analysis was carried out by counting at least 5000 spermatozoa in the excitation range of 519–630 nm in flow cytometry. Spermatozoa with green fluorescence excitation but not red fluorescence excitation (CFDA+, PI-) were identified. These spermatozoa were considered to be “viable and have an intact plasma membrane”.

The JC-1 (5,50,6,60-tetrachloro-1,10,3,30-tetraethyl benzimidazolyl-carbocyanine iodide) staining procedure was used to assess the sperm mitochondrial membrane potential (MMP) in sperm [45]. Spermatozoa exhibiting orange fluorescence excitation in flow cytometry were considered to have “high mitochondrial membrane potential (hMMP)”.

Superoxide radicals were quantitatively measured to assess ROS in spermatozoa under oxidative stress. Measurements were performed using quantitative measurement of superoxide radicals in spermatozoa under oxidative stress. The ROS+ in the thawed semen samples was measured using a Muse^®^ Oxidative Stress Kit (Luminex, Austin, TX, USA) and the manufacturer’s protocol. The semen in two straws was thawed at 37 °C for 30 s, pooled, and diluted to 100 × 10^6^ sperm mL^−1^. Subsequently, 190 μL of the kit solution was added to 10 μL of the sample. Prepared samples were placed in 96-well microplates and examined with flow cytometry. The percentage of cells with fluorescence excitation in red (ROS was detected).

### 2.5. Statistical Analysis

The experiments in this study were performed ten times (*n* = 10). SPSS Version 22.0 for Windows (SPSS Inc., Chicago, IL, USA) was used to evaluate all spermatological parameters. The normality of the data was confirmed before the analysis. One-way analysis of variance (ANOVA) was used in the statistical analysis of spermatological examination results, followed by Duncan’s Multiple Range Test for post hoc comparison of means (Appendix A). The results are presented as mean ± standard error (SE). Differences with values of *p* < 0.05 were considered statistically significant.

## 3. Results

### 3.1. CASA Results

After equilibration, the RES10 and RES20 groups containing resveratrol and the RLC10, RLCS20, and RLC40 groups containing the resveratrol + cyclodextrin complex were found to be statistically higher than the control group in terms of total motility percentage (*p* < 0.05). There was no difference in total motility values between the experimental groups (except for the control) containing chemical substances (*p* > 0.05). The progressive motility (%) and kinetic velocity parameters of all groups were similar after equilibration (*p* > 0.05) (Table 2).

The highest total motility rate after thawing was found in the RES20 and RLC10 groups, and these groups were found to be more successful than the CD40 group (*p* < 0.05). The highest progressive motility rate after thawing was detected in the RLC10 group, and the progressive motility rates in the RES10, CD10, CD40, RLC20, and RLC40 groups were found to be lower than in the RLC10 group (*p* < 0.05). The VAP value of the RLC10 group was higher compared to that of the RES10 group after thawing (*p* < 0.05). The values for VSL, VCL, ALH, BCF, STR, and LIN were similar across all groups (*p* > 0.05) (Table 3).

### 3.2. Capacitation Status, HOST, and Flow Cytometry Results

After thawing, in the fluorescence examination using CTC dye, the lowest F pattern (non-capacitated spermatozoa) was observed in the CD groups (*p* < 0.05). The proportion of F pattern spermatozoa was statistically higher in the RES20 group compared to all CD groups, as well as the control, RES40, and RLC40 groups (*p* < 0.05).

In terms of post-thaw HOST positive rates, higher rates were detected in the resveratrol-containing groups (except RES40) compared to the other groups (*p* < 0.05). The ratio of viable/membrane-intact spermatozoa determined via CFDA/PI staining was found to be similar in the control group and the RES and RLC groups (*p* > 0.05). Upon comparing the RES and RLC groups, we found that the RLC10 group had a higher ratio than the RES40 group (*p* < 0.05). After thawing, the viable sperm with intact acrosomes, as determined via flow cytometry, showed that the RES10, RES20, RES40, RLC20, and RLC40 groups had higher rates compared to the CD40 group (*p* < 0.05). However, no statistically significant difference was detected between the control group, which had a 30.95 ± 1.68% viable/intact acrosome spermatozoon rate, and the other groups (*p* > 0.05). It was determined that the CD20 and CD40 groups, which contained high levels of cyclodextrin, exhibited higher mitochondrial activity after thawing compared to the other groups (*p* < 0.05). All groups were found to be similar in terms of spermatozoon rates under oxidative stress after thawing (*p* > 0.05) (Table 4).

## 4. Discussion

Resveratrol exhibits antioxidant properties through both extracellular and intracellular mechanisms. Extracellularly, it has been reported to inhibit superoxide anions, scavenge free radicals, and donate protons. Intracellularly, it has been reported to reduce lipid peroxidation of the sperm plasma membrane as well as protect sperm proteins, mitochondria, and DNA from damage during freezing and thawing [46]. In this study, we investigated the effects of different doses of resveratrol (10, 20, and 40 µg/mL) in its pure form, as well as its cyclodextrin-loaded forms, on sperm functions during the cryopreservation of ram semen.

Following the equilibration period, the motility values of all experimental groups, with the exception of the RES40 and CD groups, exhibited statistically significant increases compared to the control group (*p* < 0.05). This indicates the positive effects of resveratrol and its complex with cyclodextrin contained in the extender, on the viability of spermatozoa during the pre-freezing phase. In this study, we hypothesize that the increase observed during the pre-freezing phase depends on both the dosage and the application method of antioxidant substances. Specifically, this improvement is attributed to the direct treatment of the antioxidant substance with semen followed by a 15-min incubation period. The absence of a statistically significant difference in progressive motility rates between groups during the post-equilibration period is consistent with the findings reported in the studies by Sarlos et al. [23] and Al-Mutary et al. [7].

Additionally, Falchi et al. [15] reported that while total motility and progressive motility rates remained similar to the control group after equilibration at 4 °C in deer semen diluted with a medium containing 10, 25, and 50 µM resveratrol, there was a notable decrease in the average path velocity (ALH value) with an increasing resveratrol dose. The average lateral head displacement, defined as the lateral movement of the sperm head, is known to increase in spermatozoa undergoing hyperactivation due to premature capacitation, a condition considered undesirable [47]. Contrary to the findings of the present study, possible explanations for the effect of resveratrol on ALH value in deer semen during the pre-freezing period may be attributed to variations in species, diluent composition, resveratrol dosage, and the procedures conducted up to the pre-freezing stage. There may be differences, especially in centrifugation and semen washing protocols.

Motility and kinetic velocity parameter values provide important information about the movement capacity of the spermatozoa required to reach the fertilization site [48]. In this study, all kinetic velocity parameters (VAP, VSL, VCL, ALH, BCF, STR, and LIN) and motility percentages of the experimental groups after thawing were determined to be statistically similar to those of the control group (*p* > 0.05). Several studies conducted across different species have demonstrated a favorable impact of resveratrol on post-thaw spermatozoa, specifically on total motility and/or progressive motility [5,8,12,15,19,22,27]. Silva et al. [26] and Brair et al. [10] observed that resveratrol did not yield an improvement in post-thaw motility and progressive motility rates in ram semen. Similarly, Langobardi et al. [21] reported similar findings in buffalo semen. Interestingly, to the contrary, Sharafi et al. [25] stated that resveratrol improved sperm motility in bulls with poor cryotolerance but was not effective in those with good cryotolerance. When evaluating kinetic velocity parameters, the results observed after thawing were consistent with the freezing study on deer semen by Falchi et al. [15]. Similarly, Shabani Nashtaei et al. [24] reported that resveratrol did not have any effect on kinetic velocity values after thawing in human semen.

However, in studies involving bull semen, Bucak et al. [12] found that 1 mM of resveratrol increased the ALH value but had no discernible effect on other velocity values. Assunção et al. [8] observed that a concentration of 0.05 mM resveratrol resulted in increased values for VSL, VAP, BCF, LIN, and STR after thawing, with no significant effect on VCL values. However, these values were negatively affected by higher doses (0.1 and 1 mM). Differences in results regarding total motility, progressive motility, and kinetic velocity values among studies may be attributable to differences in the type and dosage of resveratrol used. Additionally, differences in the specific treatments to which semen was subjected and the types of diluents used across studies may also contribute to the observed disparities in results.

Studies have explored the effect of cyclodextrin added to the diluent on sperm motility. Mocé et al. [49] reported that cyclodextrin had a detrimental effect on total motility and progressive motility rates in frozen–thawed ram semen. They attributed this to an increased sensitivity of the cells to cold shock due to the loss of membrane cholesterol. Supporting this issue, this study determined that the groups containing only CD had a proportionally lower percentage of total and progressive motility compared to the control group (*p* > 0.05). In contrast, Benhenia et al. [36] reported a positive impact of methyl-β-cyclodextrin on motility and kinetic movement parameters (VAP, VSL, and VCL) in thawed ram epididymal semen. Likewise, Zeng and Terada [50,51] found that both methyl-β-cyclodextrin and 2-hydroxypropyl-β-cyclodextrin improved motility parameters in thawed pig semen. The supplementation of cyclodextrins to the extender induces a partial reduction in cholesterol content within the membrane structure of spermatozoa [52].

Some researchers suggest that increased membrane fluidity and permeability resulting from cholesterol loss may lead to reduced formation of intracellular ice crystals during freezing. As a result, spermatozoa may potentially experience less damage during the freezing–thawing process [53]. In particular, differences in research results across studies conducted on the same species may be due to factors such as the type of cyclodextrin used, dosage administered, duration of semen treatment, and the composition of the diluent [36,54,55].

Remarkably, following the thawing process, the RLC10 group, which contained an equivalent quantity of resveratrol, exhibited a significantly improved progressive motility rate compared to the RES10 group (*p* < 0.05). This observation suggests that the inclusion of 10 µg/mL resveratrol loaded into cyclodextrin exerts a positive influence on sperm movement. In contrast, analogous improvements were not observed between the RES20 and RES40 groups and their respective cyclodextrin-loaded counterparts, namely the RLC20 and RLC40 groups. Cyclodextrins can efficiently encapsulate approximately 30–40% of resveratrol compounds due to their structural characteristics [56]. Cyclodextrins, which do not contain loaded resveratrol in their internal cavity, show affinity to the surrounding lipid structures, especially for cholesterol in the membrane structure of spermatozoa [57]. Therefore, the elevated concentration of free cyclodextrin in the RLC20 and RLC40 groups may have induced a certain degree of adverse influence on the membranes of spermatozoa.

Compared with pure resveratrol within the RES10 group, the resveratrol loaded into cyclodextrin within the RLC10 group showed higher solubility, a longer biological half-life, improved chemical stability, less sensitivity to light-induced and oxidative effects, and better interaction with the spermatozoa membrane structure after increased activity [58]. Consequently, despite the cyclodextrin-induced reduction in membrane cholesterol within the RLC10 group, the extent of this reduction was relatively modest when juxtaposed with the RLC20 and RLC40 groups. Furthermore, the RLC10 group may have exhibited greater efficacy as an antioxidant compared to the RES10 group. Therefore, it can be concluded that higher values observed in progressive motility rate and spermatozoa velocity can be attributed to these positive characteristics.

In the present study, no differences were found between the resveratrol-containing groups (RES and RLC) and the control group in terms of the ratio of viable/membrane-intact spermatozoon. Consistent with the present study, Silva et al. [26] reported that 5, 10, 15, and 20 µg/mL resveratrol used in ram semen cryopreservation did not affect sperm membrane integrity after thawing. Similarly, in another study, 50 μm/mL resveratrol was not effective in preserving sperm plasma membrane integrity after thawing in rams [10]. Conversely, it has been reported that resveratrol is effective in maintaining membrane integrity after thawing in studies conducted in pig [27] and deer [15] semen. In the assessment of functional membrane integrity using the hypo-osmotic swelling test (HOST), significantly elevated rates were observed in the RES10, RES20, and RLC groups (*p* < 0.05). It was reported that 50 and 100 µM resveratrol increased the HOST positive spermatozoon ratio in buffalo bull semen after thawing [5]. Notably, in the RES40 group, the HOST-positive ratio was comparatively lower than in the other RES groups, and a similar trend, though not statistically significant, was observed in the ratio of viable/membrane-intact spermatozoa. This observation suggests a potential adverse impact of a high dose (40 µg/mL) of resveratrol on the structural integrity of spermatozoa membranes. Consistent with the current study, Lv et al. [22] documented that 10 and 50 µM resveratrol concentrations preserved functional membrane integrity in goat semen. However, this positive effect underwent a negative alteration with increasing dosage. The higher HOST (+) rates observed in the study compared to the rates obtained with CFDA-PI staining may be attributed to the differences in the methodologies. While HOST identifies spermatozoa that are biochemically active and sensitive to a hypo-osmotic environment, the vital dye CFDA has the capacity to detect microphysical damage. Additionally, centrifugation and washing of the semen during the CFDA staining protocol could potentially contribute to a decrease in the proportion of spermatozoa with intact membranes.

Several investigations have been conducted to assess the impact of resveratrol on acrosome integrity in the preservation of semen from various animal species. Certain studies, such as those conducted by Bucak et al. [12], Gadani et al. [16], Alhelal et al. [6], and Kaeoket and Chanapiwat [19], reported a positive result on acrosome integrity. Conversely, other studies, including those by Bang et al. [9], Lv et al. [22], Ahmed et al. [5], Zhu et al. [27], Assunção et al. [8], and Li et al. [20], demonstrated no significant effects. In ram semen, according to the findings presented by Silva et al. [26], the application of resveratrol during the freezing process did not have a noteworthy impact on the rates of acrosome integrity in post-thawed semen. In contrast, Sarlos et al. [23] reported a notable reduction in acrosome damage rates when resveratrol was employed during short-term cold storage. In the present study no observed positive effects of resveratrol on acrosome integrity rates were obtained after the thawing process. Despite the alignment of our findings with the study conducted by Silva et al. [26] on ram semen, it is noteworthy that the proportion of spermatozoa exhibiting intact acrosomes after thawing was lower in the groups containing resveratrol. The probable reason for this is that although the fluorescent stain used in both studies was the same, the examination method was different. The researchers used phase-contrast microscopy with fluorescent attachment for the examination of acrosome integrity. In the present study, flow cytometry, which can give more sensitive results, was used.

During the cryopreservation process, changes in the phase state of the lipid membrane of ram spermatozoa, specifically the transition from the liquid phase to the gel phase, result in approximately 14% depletion of cholesterol from the plasma membrane [49]. This loss disrupts the membrane fluidity of spermatozoa, causing temporary pores to form. Capacitation-like changes occur due to increased Ca^++^ entry into the intracellular environment through these pores. This condition, which develops due to cold shock in the in vitro environment, is called “cryo-capacitation” and is a primary factor that causes a decrease in the fertilization ability of spermatozoa [59,60]. Studies have documented a positive correlation between elevated intracellular calcium ion concentration and the requisite protein tyrosine phosphorylation for processes such as capacitation and acrosome reactions [61,62,63]. Although the mechanism has not been fully elucidated, it has been reported that resveratrol inhibits early capacitation in buffalo bull spermatozoa [21]. The potential rationale for the inhibition of capacitation-like changes via resveratrol are believed to include a reduction in ROS within the environment and a decrease in protein tyrosine phosphorylation, a process regulated in the presence of ROS [21]. Additionally, in a study conducted with rat ventricular muscle cells, it was reported that resveratrol contributed to the reduction in intracellular free calcium ion concentration by inhibiting cell membrane calcium channels. This action resulted in a reduction in the activity of the tyrosine kinase enzyme, consequently inhibiting protein tyrosine phosphorylation [64]. Desroches et al. [65] reported that quarcetin, a polyphenol compound similar to resveratrol, inhibited capacitation “reversibly”; Longobardi et al. [21] reported that resveratrol, which decreased the early capacitation rate after thawing in buffaloes, did not adversely affect in vitro fertilization rates. Similarly, Brair et al. [10] showed that 50 µM/mL resveratrol improved fertilization capacity by preventing pre-capacitation in ram spermatozoa. The findings in the present study are consistent with the studies of Longobardi et al. [21] and Brair et al. [10], which found that 20 µg/mL resveratrol in the semen extender inhibited cryo-capacitation in ram spermatozoa (*p* < 0.05). Conversely, CDs show a high affinity for steroids and may function as agents to induce sperm capacitation in vitro by facilitating the removal of cholesterol from the membrane structure [52]. This mechanism accounts for the significantly lower F pattern sperm rates observed in the CD groups compared to the control group in this study.

The continuity of mitochondrial activity is essential for spermatozoon motility. However, high mitochondrial activity causes an increase in reactive oxygen species due to oxidative phosphorylation [66] and leads to the premature depletion of sperm energy resources, thereby shortening the lifespan of the spermatozoon [67]. Resveratrol, when added to the extender in the freezing of bull [12], buffalo bull [5], pig [27], billy goat [22], and rooster [68] semen, has been reported to increase spermatozoon mitochondrial activity after thawing. Conversely, it has been reported to negatively affect mitochondrial activity in ram semen [26]. In this study, the groups CD20 and CD40, which contained higher amounts of cyclodextrin, exhibited the highest rates of mitochondrial activity (*p* < 0.05). The primary energy production pathway of spermatozoa is mitochondrial oxidative phosphorylation, and its regulation occurs in the presence of calcium [69]. Considering the spermatozoa membrane integrity, acrosome integrity, and cryo-capacitation rates in these groups suggests that cyclodextrin-induced membrane cholesterol loss alters membrane structure and increases intracellular calcium influx, thereby enhancing mitochondrial activity. The lower mitochondrial activity observed in the CD10 group can be attributed to its lower cyclodextrin content. In the RES20 group, where early capacitation was prevented, no significant difference was observed compared to the control group, despite the expectation of decreased mitochondrial activity due to reduced intracellular calcium entry. A similar result was detected in the study conducted by Brair et al. [10] using 50 μm/mL resveratrol. However, although not statistically significant, it was noted that mitochondrial activity decreased with increasing doses in the RES and RLC groups containing resveratrol.

The method used to determine the severity of oxidative stress in this study is based on the detection of intracellular ROS using dihydroethidium. Dihydroethidium is a dye specifically used for the detection of the superoxide radical (O_2_) [70]. In this study, the absence of a statistically significant difference between the groups indicates that resveratrol at concentrations of 10, 20, and 40 µg/mL had no effect on intracellular O_2_ radicals in ram spermatozoa. Longobardi et al. [21,71] reported in their research that the antioxidant substances (carnitine and resveratrol) used in the freezing of buffalo sperm reduced intracellular O_2_ radicals but were not effective in lipid peroxidation (LPO). To the contrary, a study on ram epididymal sperm by Merati et al. [72] reported that although minocycline did not affect the intracellular O_2_ rate after thawing, it was effective in reducing the amount of intracellular H_2_O_2_ and malondialdehyde (MDA), the end product of membrane LPO. In the present study, when oxidative stress values were evaluated alongside other spermatological findings (particularly HOST, partial membrane, and acrosome integrity), it was concluded that the addition of resveratrol is beneficial. This indicates that resveratrol has a protective effect on the spermatozoon membrane structure. Similar studies on bull [12], buffalo bull [1], deer [15], and boar [19] sperm have reported that resveratrol reduces the amount of MDA. For this reason, it is recommended to measure MDA levels in studies involving frozen ram sperm to determine the effectiveness of resveratrol on LPO rather than focusing solely on intracellular oxidative stress.

## 5. Conclusions

Overall, varying doses of resveratrol in frozen ram semen did not statistically improve post-thaw motility or other kinematic parameters related to movement. However, resveratrol at a concentration of 20 µg/mL in the extender was effective in suppressing early post-thaw capacitation. Additionally, low doses of resveratrol (10 and 20 µg/mL) were observed to preserve the functional integrity of the plasma membrane. In contrast, the resveratrol–cyclodextrin complex did not affect motility or kinematic parameters. Despite preserving the functional integrity of the plasma membrane, its ability to suppress early capacitation was limited. In conclusion, both resveratrol and the resveratrol–cyclodextrin complex were found to be beneficial for the cryopreservation of ram sperm, although higher concentrations of resveratrol negatively affected sperm functions.

## Figures and Tables

**Figure 1 animals-14-02745-f001:**
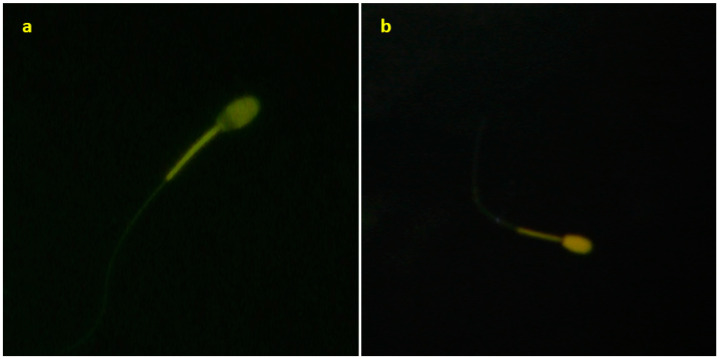
Sperm images obtained with CTC fluorescence staining: (**a**) B pattern; (**b**) F pattern.

**Figure 2 animals-14-02745-f002:**
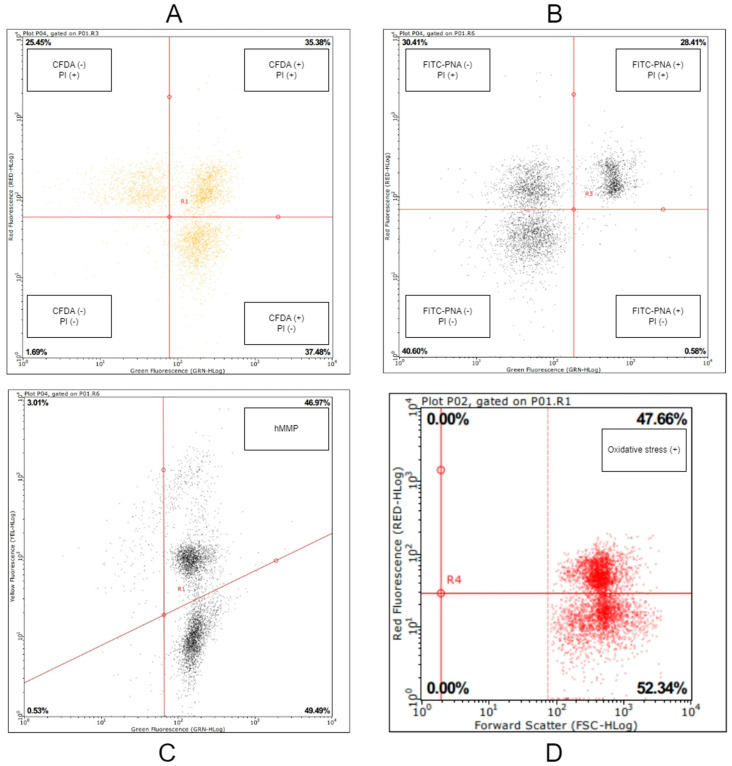
Sample data plot graphs of flow cytometry analyses: (**A**) membrane integrity, (**B**) acrosome integrity, (**C**) high mitochondrial activity, and (**D**) oxidative stress.

**Table 1 animals-14-02745-t001:** Experimental groups and treatment details.

Groups	Treatment
**Control**	TCG + TEY
**RES10**	TCG containing 10 µg/mL RES (in total volume) + TEY
**RES20**	TCG containing 20 µg/mL RES (in total volume) + TEY
**RES40**	TCG containing 40 µg/mL RES (in total volume) + TEY
**CD10**	TCG containing 178.65 µg/mL CD (in total volume) + TEY
**CD20**	TCG containing 357.30 µg/mL CD (in total volume) + TEY
**CD40**	TCG containing 714.60 µg/mL CD (in total volume) + TEY
**RLC10**	TCG containing 10 µg/mL RES loaded CD (10 µg/mL RES + 178.65 µg/mL CD in total volume) + TEY
**RLC20**	TCG containing 20 µg/mL RES loaded CD (20 µg/mL RES + 357.30 µg/mL CD in total volume) + TEY
**RLC40**	TCG containing 40 µg/mL RES loaded CD (40 µg/mL RES 714.60 µg/mL CD in total volume) + TEY

RES: Resveratrol; CD: Cyclodextrin; RLC: Resveratrol-loaded Cyclodextrin; TCG: Tris–citric acid–glucose medium; TEY: Tris-based egg yolk extender.

**Table 2 animals-14-02745-t002:** Means ± SE of sperm motility characteristics of post-equilibration ram sperm across experimental groups (*n* = 10).

Group	CASA								
	Total Motility (%)	Progressive Motility (%)	Kinetic Velocity Parameters						
			VAP (μm/s)	VSL (μm/s)	VCL (μm/s)	ALH (μm/s)	BCF (Hz)	STR (%)	LIN (%)
**Control**	91.90 ± 1.27 ^b^	51.20 ± 2.99	122.58 ± 5.27	102.25 ± 4.92	191.85 ± 9.09	6.56 ± 0.28	35.93 ± 1.14	82.10 ± 1.33	56.10 ± 1.98
**RES10**	96.20 ± 0.64 ^a^	54.20 ± 3.00	124.87 ± 4.00	104.03 ± 3.63	191.78 ± 8.26	6.54 ± 0.33	34.54 ± 1.39	82.40 ± 1.60	57.00 ± 2.48
**RES20**	96.00 ± 0.51 ^a^	52.00 ± 3.23	123.91 ± 4.17	102.02 ± 3.84	192.49 ± 8.25	6.64 ± 0.40	34.78 ± 1.17	81.60 ± 1.51	55.90 ± 2.28
**RES40**	94.10 ± 1.40 ^ab^	51.90 ± 2.96	123.37 ± 3.70	101.60 ± 3.56	192.09 ± 8.06	6.54 ± 0.35	33.84 ± 1.28	81.40 ± 1.58	55.50 ± 2.68
**CD10**	93.30 ± 0.96 ^ab^	50.60 ± 2.54	122.38 ± 3.53	102.01 ± 3.15	189.46 ± 6.44	6.40 ± 0.28	35.38 ± 1.27	82.40 ± 1.38	56.40 ± 2.14
**CD20**	93.30 ± 1.35 ^ab^	51.80 ± 3.15	123.26 ± 3.89	103.25 ± 3.94	189.74 ± 6.77	6.42 ± 0.35	35.54 ± 1.42	82.50 ± 1.68	56.60 ± 2.67
**CD40**	93.40 ± 0.74 ^ab^	53.50 ± 2.67	126.23 ± 3.86	105.98 ± 4.12	192.93 ± 5.24	6.44 ± 0.29	34.85 ± 1.31	82.70 ± 1.31	57.10 ± 2.30
**RLC10**	95.30 ± 0.47 ^a^	50.90 ± 3.50	124.97 ± 4.63	102.46 ± 5.09	191.99 ± 6.23	6.52 ± 0.31	33.05 ± 1.37	80.90 ± 1.68	55.50 ± 2.65
**RLC20**	95.40 ± 0.68 ^a^	53.50 ± 3.00	126.63 ± 4.27	105.05 ± 4.25	190.88 ± 8.14	6.18 ± 0.37	33.64 ± 1.05	82.60 ± 1.60	58.30 ± 2.81
**RLC40**	95.20 ± 0.69 ^a^	52.10 ± 2.97	126.96 ± 5.16	104.56 ± 4.57	192.44 ± 9.53	6.36 ± 0.34	34.20 ± 0.81	82.00 ± 1.57	57.70 ± 2.75

Within columns, means with no common letters are statistically different (^ab^: *p* < 0.05). VAP; average pathway velocity, VSL; velocity straight line, VCL; curvilinear velocity, ALH; amplitude of lateral head displacement, BCF; beat cross frequency, STR; straightness, LIN; linearity, RES10; 10 µg/mL resveratrol, RES20; 20 µg/mL resveratrol, RES40; 40 µg/mL resveratrol, CD10; 178.65 µg/mL cyclodextrin, CD20; 357.30 µg/mL cyclodextrin, CD40; 714.60 µg/mL cyclodextrin, RLC10; 10 µg/mL resveratrol-loaded cyclodextrin, RLC20; RLC20; 20 µg/mL resveratrol-loaded cyclodextrin, RLC40; 40 µg/mL resveratrol-loaded cyclodextrin.

**Table 3 animals-14-02745-t003:** Mean ± SE of sperm motility characteristics of post-thaw ram sperm among experimental groups (*n* = 20).

Group	CASA								
	Total Motility (%)	Progressive Motility (%)	Kinetic Velocity Parameters						
			VAP (μm/s)	VSL (μm/s)	VCL (μm/s)	ALH (μm/s)	BCF (Hz)	STR (%)	LIN (%)
**Control**	59.65 ± 3.66 ^ab^	29.15 ± 1.59 ^ab^	95.35 ± 2.84 ^ab^	84.84 ± 3.14	146.84 ± 3.39	5.79 ± 0.13	37.15 ± 0.71	85.85 ± 0.85	57.75 ± 1.16
**RES10**	57.25 ± 3.07 ^ab^	26.30 ± 1.90 ^b^	90.52 ± 2.27 ^b^	79.78 ± 2.34	140.77 ± 3.38	5.99 ± 0.16	34.97 ± 0.64	84.95 ± 0.78	56.70 ± 1.08
**RES20**	62.65 ± 3.26 ^a^	30.45 ± 1.23 ^ab^	92.74 ± 1.92 ^ab^	82.14 ± 2.30	142.27 ± 2.66	5.81 ± 0.20	35.69 ± 0.72	85.90 ± 0.94	58.05 ± 1.46
**RES40**	60.90 ± 2.56 ^ab^	28.90 ± 1.60 ^ab^	93.10 ± 2.16 ^ab^	82.08 ± 2.51	143.65 ± 2.90	5.90 ± 0.21	35.36 ± 0.75	85.10 ± 1.06	57.15 ± 1.59
**CD10**	57.85 ± 3.60 ^ab^	26.75 ± 1.63 ^b^	92.37 ± 1.99 ^ab^	81.45 ± 2.56	142.25 ± 2.24	5.85 ± 0.19	36.27 ± 0.89	85.25 ± 1.12	57.40 ± 1.67
**CD20**	59.80 ± 3.42 ^ab^	28.90 ± 1.55 ^ab^	95.43 ± 2.54 ^ab^	85.02 ± 2.73	149.05 ± 3.90	5.90 ± 0.16	37.39 ± 0.60	85.60 ± 0.89	56.80 ± 1.23
**CD40**	51.50 ± 3.70 ^b^	26.20 ± 2.24 ^b^	95.50 ± 1.94 ^ab^	85.21 ± 2.28	147.14 ± 2.77	5.75 ± 0.18	36.89 ± 0.70	86.10 ± 0.92	58.00 ± 1.37
**RLC10**	64.35 ± 2.64 ^a^	32.65 ± 1.44 ^a^	98.15 ± 1.72 ^a^	87.61 ± 2.00	148.10 ± 2.84	5.70 ± 0.18	37.25 ± 0.52	85.95 ± 0.87	59.25 ± 1.62
**RLC20**	56.85 ± 2.96 ^ab^	26.70 ± 1.08 ^b^	94.61 ± 2.59 ^ab^	83.52 ± 2.98	146.18 ± 3.01	5.96 ± 0.17	35.42 ± 0.84	85.25 ± 1.09	57.30 ± 1.55
**RLC40**	57.00 ± 3.20 ^ab^	26.55 ± 1.64 ^b^	92.78 ± 2.35 ^ab^	81.12 ± 2.92	143.71 ± 2.64	5.94 ± 0.20	35.42 ± 1.02	84.05 ± 1.21	56.20 ± 1.53

Within columns, means with no common letters are statistically different (^ab^: *p* < 0.05). VAP; average pathway velocity, VSL; velocity straight line, VCL; curvilinear velocity, ALH; amplitude of lateral head displacement, BCF; beat cross frequency, STR; straightness, LIN; linearity, RES10; 10 µg/mL resveratrol, RES20; 20 µg/mL resveratrol, RES40; 40 µg/mL resveratrol, CD10; 178.65 µg/mL cyclodextrin, CD20; 357.30 µg/mL cyclodextrin, CD40; 714.60 µg/mL cyclodextrin, RLC10; 10 µg/mL resveratrol-loaded cyclodextrin, RLC20; RLC20; 20 µg/mL resveratrol-loaded cyclodextrin, RLC40; 40 µg/mL resveratrol-loaded cyclodextrin.

**Table 4 animals-14-02745-t004:** Mean ± SE of F pattern, HOST +, intact plasma membrane, intact acrosome, high mitochondrial activity, and oxidative stress status of frozen–thawed ram sperm among groups (*n* =10).

Group	F Pattern (%)	HOST+ (%)	Plasma Membrane Integrity (%)	Acrosome Integrity (%)	High Mitochondrial Activity (%)	Oxidative Stress (%)
**Control**	44.00 ± 1.94 ^b^	35.15 ± 1.03 ^b^	30.03 ± 1.87 ^abc^	30.95 ± 1.68 ^ab^	55.92 ± 2.73 ^b^	38.88 ± 1.63
**RES10**	48.40 ± 2.64 ^b^	43.80 ± 1.40 ^a^	34.81 ± 2.61 ^ab^	32.43 ± 2.01 ^a^	55.22 ± 2.63 ^b^	37.63 ± 1.58
**RES20**	59.10 ± 2.08 ^a^	44.65 ± 1.11 ^a^	30.83 ± 2.37 ^abc^	33.70 ± 1.60 ^a^	53.10 ± 2.89 ^b^	37.66 ± 1.95
**RES40**	46.80 ± 2.57 ^b^	37.45 ± 1.39 ^b^	28.82 ± 2.05 ^bcd^	33.55 ± 1.68 ^a^	52.89 ± 2.23 ^b^	39.49 ± 1.98
**CD10**	31.00 ± 2.45 ^c^	36.00 ± 1.40 ^b^	31.88 ± 2.00 ^abc^	29.58 ± 1.87 ^ab^	55.44 ± 3.10 ^b^	39.17 ± 1.47
**CD20**	29.70 ± 2.99 ^c^	33.60 ± 1.38 ^b^	25.63 ± 1.65 ^cd^	28.78 ± 1.44 ^ab^	64.76 ± 1.51 ^a^	38.91 ± 1.35
**CD40**	25.30 ± 2.74 ^c^	33.60 ± 1.75 ^b^	23.45 ± 1.86 ^d^	25.94 ± 1.47 ^b^	65.77 ± 2.89 ^a^	38.79 ± 1.27
**RLC10**	52.40 ± 4.05 ^ab^	41.40 ± 1.53 ^a^	36.43 ± 1.39 ^a^	29.30 ± 2.80 ^ab^	54.52 ± 2.97 ^b^	36.96 ± 2.30
**RLC20**	51.20 ± 2.59 ^ab^	43.25 ± 1.23 ^a^	34.67 ± 1.79 ^ab^	34.01 ± 1.79 ^a^	49.57 ± 2.60 ^b^	37.18 ± 1.71
**RLC40**	46.10 ± 2.72 ^b^	42.95 ± 1.25 ^a^	33.48 ± 2.67 ^ab^	32.86 ± 2.76 ^a^	48.33 ± 1.99 ^b^	37.70 ± 1.67

Within columns, means with no common letters are statistically different (^abcd^: *p* < 0.05). RES10; 10 µg/mL resveratrol, RES20; 20 µg/mL resveratrol, RES40; 40 µg/mL resveratrol, CD10; 178.65 µg/mL cyclodextrin, CD20; 357.30 µg/mL cyclodextrin, CD40; 714.60 µg/mL cyclodextrin, RLC10; 10 µg/mL resveratrol-loaded cyclodextrin, RLC20; RLC20; 20 µg/mL resveratrol-loaded cyclodextrin, RLC40; 40 µg/mL resveratrol-loaded cyclodextrin.

## Data Availability

All data generated or analyzed during this study are included in this article.

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
