# Peer review of "Effects of Resveratrol-Loaded Cyclodextrin on the Quality Characteristics of Ram Spermatozoa Following Cryopreservation"

_animals, 2024, doi:10.3390/ani14182745_

Round 1

Reviewer 1 Report

Comments and Suggestions for Authors

The manuscript “Effectiveness of Resveratrol-Loaded Cyclodextrin in the Cryopreservation of Ram Semen” evaluated the impact of resveratrol-loaded cyclodextrin on the quality of ram sperm after thawing. The data is interesting, and the writing is well-crafted and accurate. I suggest specific corrections according to the items in the manuscript.

Abstract and keywords:

1. I suggest two points in the abstract: a) The authors should better defend the tests with cyclodextrin; b) Present numerical values ​​of the results. Explore the results further.

2. Replace keywords that are repeated in the manuscript title.

Introduction:

1. I suggest inserting 1-2 sentences better explaining the physical-chemical characteristics of cyclodextrin.

2. Under what biological conditions has cyclodextrin been used? Explain, citing semen or other cell types.

3. What biochemical tests exist for cyclodextrin? And its combination with revesratrol?

Material and methods:

1. Present the values ​​in "g" for 250 rpm.

2. How were the concentrations of the experimental groups chosen? Report this in materials and methods.

Results and discussion:

1. Explain the role of revestrarol and cyclodextrin in sperm capacitation.

2. The caption for Figure 1 needs to be improved.

Reviewer 2 Report

Comments and Suggestions for Authors

In this study the Authors investigated the effects of different concentrations of resveratrol, either in pure form or loaded into methyl-β-cyclodextrin, on ram sperm quality characteristics following cryopreservation. The Reviewer suggests that the following comments would be helpful to improve the quality of the manuscript.

Comments are as follows:

1.Title. It should be re-phrased - possible suggestion "Effects of resveratrol-loaded cyclodextrin on the quality characteristics of ram spermatozoa following cryopreservation"

2. Abstract. Should consider to re-write it .

a) For examples, sperm quality characteristics evaluated included CASA-analyzed motility and motion parameters, membrane integrity, capacitation status, etc.

b) It should be  "…..adversely affected sperm functions following freezing-thawing"  (L32).

c) Should consider to replace "spermatological properties" with "sperm functions" throughout the text, where it is appropriate (L32).

d) L80 "…. on  ram sperm quality characteristics following cryopreservation"

3. M&M

a) L118-126. To improve clarity provide a table showing the treatment procedure. In its present description, the procedure is very difficult to follow. Define "RES", "CD", etc -L104.

b) 2.4. "Sperm quality assessment"

c) Missing is the "amplitude of lateral head displacement (ALH, mm)".

d) Give the duration of the post-equilibration period (L142), and in the caption of Table 1.

e) Why two plasma membrane integrity tests were performed - HOS test and CFDA/PI? Their mechanisms are different (L369-374), but they are still useful membrane integrity tests, and in most cases, the results are correlated.

f) For the CTC test, a third fluorescence pattern is missing (L150-160). Why the acrosome-reacted sperm population was not recorded in this study. Should consider to present all 3 fluorescence patterns (3 sperm populations are analyzed by the CTC test) in Table 3.

g) 2.4.4. possible suggestion - "Flow cytometric analyses of sperm membrane integrity, mitochondrial functions and oxidative stress status"

h) Re-write the statement in L172-174. L180 -"….to  assess the sperm mitochondrial membrane potential, MMP" [42].

4. Statistical analysis (L195-200)

Incomplete analysis makes it very difficult to interpret the acquired data.

i) Were the data checked with the normality test to validate the use of  parametric tests (ANOVA, Duncan etc)?

ii) Velocity parameters, ALH and BCF, sometimes require log-transformation prior to ANOVA analysis. Did these data follow the normal distribution patterns? If not, then use a nonparametric ANOVA test, for example, the Kruskal-Willis test.

iii) Provide the ANOVA results as Supplementary materials.

5.Results/Tables

a) Should consider to re-run the statistical analysis.

b) The caption of each table needs to be improved. Give the explanation of the treatment procedure, for example, what is RES10 referring to? Provide the number of analyzed samples. Re-check the table headings, improve the visibility of the significant levels, etc.

6. Discussion

a) Should consider to reduce the Discussion. For example, L394-419.

b) MDA -malondialdehyde (L453)

Comments on the Quality of English Language

Some English corrections are required throughout the text.

Round 2

Reviewer 2 Report

Comments and Suggestions for Authors

The Authors have satisfactorily addressed all of my comments.